# Period Poverty from a Public Health and Legislative Perspective

**DOI:** 10.3390/ijerph20237118

**Published:** 2023-11-28

**Authors:** Simranjot Mann, Sharon K. Byrne

**Affiliations:** The College of New Jersey, The School of Nursing and Health Sciences, Ewing Township, NJ 08618, USA; byrnes@tcnj.edu

**Keywords:** period poverty, public health crisis, reproductive justice, policy, legislation, menstruation equity, menstruation management, menstruation hygiene, gender

## Abstract

Period poverty is a global issue that needs to be addressed as a public health crisis. It is directly related to Sustainable Health Goals three, four, five, six, and eight. Period poverty adversely affects the health of anyone capable of menstruating, which is nearly half of the world population, at the physiological, emotional, and psychosocial level. Biases, cultural beliefs, ethical reproductive justice issues, social stigma, and systemic factors contribute to period poverty. Every month, certain menstruators are disproportionately impacted by period poverty and struggle to access basic hygienic necessities. Important stakeholders include not only the individual who experiences menses but also educators and school systems, healthcare professionals, policymakers, public health officials, and researchers. Everyone has a role in addressing period poverty by voting for officials that proactively support legislation, policy, and programs at all levels to effectively advocate for menstrual equity and address barriers contributing to period poverty. This includes policies that increase access to menstrual hygiene products, safe menstrual management methods, and reproductive and women’s health education. Programs globally that focus on capacity building and sustainability strategies can be used as models to reduce period poverty, thereby fostering a sense of empowerment and menstruators’ sense of autonomy, dignity, and equality.

## 1. Introduction

Although the menstruating population has experienced limited access to women’s reproductive education, menstrual hygiene products, and poor menstrual hygiene management methods for decades, this overlooked public health crisis has been termed “period poverty” by present-day menstrual equity advocates. Menstruation is a physiological process that involves the shedding of the uterine lining [1]. Appropriate tools need to be used to safely collect menses to prevent infection [2]. Menstruation pads and tampons are commonly used [3]. However, globally, the menstruating population may experience inadequate access to menstrual hygiene products and sanitation facilities [3]. Menstruators may be forced to use old clothes, paper, cotton, wool, and leaves [3]. Period poverty is not limited to the lack of access to hygiene products, washing facilities, and waste management [4]. Social taboos and stigmas regarding menstruation suppress conversations and obstruct appropriate education about menstruation, contributing to misinformation, confusion, fear, and negative attitudes towards menstruation [5].

Period poverty is an important global issue that needs to be addressed as an urgent public health crisis. It is directly and indirectly related to Sustainable Development Goals (SDGs) three, four, five, six, and eight that were published in 2015 by member states of the United Nations (UN) to provide a clear framework of goals for governments and communities to integrate into policy and initiatives to support sustainable practices that will ensure a better tomorrow. The UN outlines the importance of promoting good health and wellness, as stressed by the third SDG [6]. This goal is related to the period poverty issue, as noted above, in that it can adversely affect all aspects of health. It can also create barriers for menstruators in terms of actively participating in educational and workplace settings, thus negatively impacting education, the ability to secure employment in professional settings, and financial independence, as addressed by SDGs four and eight [6]. As outlined by the fifth SDG, efforts are needed to empower women and girls to achieve gender equality [6]. Period poverty puts menstruators at a disadvantage and reinforces gender inequities, which continues a cycle of sexism and social injustice. Henceforth, addressing period poverty related to SDG five would enhance the opportunity for menstruators to participate in society with dignity and equality, much like their male counterparts. Finally, menstruators in many parts of the world need access to clean water and sanitation facilities to maintain adequate menstrual hygiene, as emphasized by the sixth SGD [6]. The lack of access to appropriate menstrual hygiene management is evidently an important health concern that universally impacts sustainability and the health of communities across the globe.

## 2. Literature Review

A current search of the literature on this health issue over the past five years was conducted using CINAHL, Nursing and Allied Health Collection: Comprehensive, Pub Med, and Google Scholar from 2018 through November 2023. Keywords used included period poverty, menstruation equity, menstruation management, menstruation hygiene, public health crisis, reproductive justice, policy, legislation, and gender. Additional filters included age range from adolescent to adult menstruators and publication in the English language. The search yielded 581 results. Following review, a total of 31 articles and websites were included as resources. The authors categorized information related to the topic of period poverty into four themes for organizational purposes: overlooked public health cisis; repercussions of period poverty on health, wellness, and education; menstrual management in a global context; and policy, accessibility to menstrual hygiene management, and menstrual equity efforts (see Table 1).

Overlooked Public Health Crisis. Period poverty affects anyone capable of menstruating, which is equivalent to nearly half of the world’s population [7,8]. Focusing on the United States, two in every five women report experiencing difficulty purchasing menstrual hygiene products [9]. Although the United States is considered one of the most developed nations in the world, period poverty is continuing to increase in the United States [10]. Between 2018 and 2021, period poverty increased by 35 percent [10]. Low- and middle-income women are disproportionately affected by period poverty [11]. In 2021, period poverty likely affected one in every six women between the ages of 12 and 44 years living below the federal poverty line (FPL) in the United States [9]. With reduced economic means to afford menstrual hygiene products, women of lower socio-economic status are more likely to not use menstrual products, use hygiene products for a longer period of time than advised, and reuse products [11]. Nearly 61 percent of the United States population reports living paycheck-to-paycheck, which suggests more than half of the population has limited financial means to purchase menstrual products [8]. 

Although there is a growing population of menstruators struggling to afford essential menstrual hygiene products, there is a lack of recognition of period poverty as a health crisis globally. Specific populations predisposed to poor menstrual hygiene management by social and systemic inequities include homeless, incarcerated, migrant, and refugee menstruators. For example, nearly 13 percent of the incarcerated population is composed of women, who depend on prison staff to address their medical and hygiene needs [12]. Inmates are often subjected to humiliation, abuse, and even rape in exchange for menstrual products by those with higher power in the prison system, including patrolling officers [11]. Although a necessity, the lack of access to menstrual products prompts incarcerated menstruators to wear unclean clothing entrenched with menses for an extended period of time until offered an opportunity to wash their clothes [11]. Another vulnerable population that experiences period poverty are those that find themselves displaced or homeless, including migrants and refugees [12]. The UN has shared that women in refugee camps and those displaced from their homes are at a greater risk of encountering period poverty [6]. Even though some members of the homeless population seek refuge in shelters, many displaced individuals do not have consistent and adequate access to sanitary products and washing facilities to maintain hand and perineal hygiene in this type of living situation [12]. Shelters provide limited living space for homeless individuals and are unable to provide consistent access to menstrual hygiene management. Additionally, menstruators may not feel safe utilizing toilets and washing facilities provided at shelters for the displaced and may view menstruation in a negative context. 

Repercussions of Period Poverty on Health, Wellness, and Education. Period poverty is correlated with adverse health and social outcomes at both the individual and societal level [2,5,13,14,15,16,17]. The lack or prolonged use of hygienic products to sanitarily collect menstrual bleeding can predispose menstruators to urogenital infections, psychosocial stress, and decreased accessibility to school and work opportunities [5]. Insufficient access to sanitary facilities, like toilets and hand washing facilities equipped with sanitary products, and hygienic waste management prevent menstruators from attending school and working [2]. In the United States, it has been estimated that one in four students either struggles to afford or is unable to afford menstrual products [17]. Approximately 84 percent of students have either missed or know someone who has not attended class due to a lack of access to menstrual products [1]. In a study by Kuhlmann et al. [16], nearly two-thirds of menstruating urban high school students in the State of Missouri reported experiencing period poverty each month [16]. Similarly, decreased class attendance among U.S. college students related to period poverty was noted to occur in 10 percent of that population [15]. The same study sample self-reported increased rates of moderate-to-severe depression due to this same issue with Latinx, Black, and first-generation college students experiencing period poverty at disproportionate rates compared to their peers [15]. Similarly, women and girls from global communities, such as Africa, are not able to attend school and work because of poor menstrual hygiene management tools [13]. In exchange for menstrual products, it has been shared that some menstruators engage in sexual activity, increasing their susceptibility to sexually transmitted infections and sexual and gender-based violence [13]. In India, the lack of private bathrooms, hygiene products, reproductive health education, and poor sanitation facilities compels 23 million adolescents to drop out of school every year as they approach puberty [18]. Dropping out of school or not being able to participate in work-related activities due to period poverty limits both the present and future financial resources of menstruators. Menstruators, regardless of their ethnic background, nationality, or race, may have to choose between purchasing food or menstrual hygiene products. Often, they compromise their hygienic needs and prioritize other necessities to survive and provide for their families [6]. Rationing or using products for a longer period of time than advised by the product manufacturers or health practitioners can also lead to adverse health conditions such as toxic shock syndrome, polycystic ovary syndrome, and endometriosis [14]. 

Not only can period poverty contribute to a decreased state of physical health, but the inaccessibility of resources and stigma associated with menses adversely affect the emotional health of many individuals and communities. This has been evidenced in the literature of menstruators experiencing increased distress scores and elevated levels of anxiety and depression [2,15]. Cultural taboos, secrecy, and embarrassment suppress discussions about menstruation, especially in rural and low- and middle-income communities throughout the globe [5]. In some cultures, menstrual stigmas and cultural beliefs prevent appropriate menstrual hygiene. For example, in some regions of Afghanistan, cultural beliefs regarding infertility prevent menstruators from touching or washing their genitals, increasing the menstruators’ susceptibility to developing urogenital infections [6]. In India and Nepal, women have been noted to be secluded from daily living activities and subject to animal attacks and sexual violence during menstruation [6]. 

Policy, Accessibility to Menstrual Hygiene Management, and Menstrual Equity. The law can be used as a tool to identify the rights of an individual, create legislation, allocate resources, impact the accessibility and quality of healthcare needs, and promote positive health outcomes [16,19]. While the lack of period products has been identified as a barrier for some students from attending school, some menstruators use and rely on the sanitary products offered by their schools, at no cost, through government funding [16]. In the spring of 2019, a member of the House of Representatives introduced the Menstrual Equity for All Act [17]. The bill (H.R. 1882) attempted to increase access to menstrual products in schools, other educational settings, incarceration facilities, shelters, employers with over 100 employees, and public federal buildings [17]. It aimed to provide menstruators with more opportunities to attend work, school, and participate in society [17]. The bill was reintroduced in May of 2021 as H.R. 3614, with some expansion, including requiring Medicaid to cover menstrual products, but was not enacted. [17]. Within the United States, as of 12 October 2022, legislation requiring schools to provide free menstrual products to students had been passed in 18 states and the District of Columbia [10]. An example of this was documented in a study of menstruators attending an urban high school in the state of Missouri. It was reported that two-thirds of the sample of menstruating high school students used period products offered by their school at least once during the school year [16]. A third rendition of the act, H.R. 3646, the Menstrual Equity for All Act of 2023, was introduced in the United States’ House of Representatives in May 2023 and referred to the Subcommittee on Health that same month [11,17]. Most recently, the state of New Jersey passed legislation in June 2023, becoming the 23rd State requiring public schools to provide menstrual products to students in grades six through twelve. At the start of academic year 2023–2024, the menstrual products, purchased by the school district and reimbursed by the state, will be available in at least half of all female and gender-neutral bathrooms [20]. Similar to the state of New Jersey and attempts by the United States’ Congress, on the global front, Scotland unanimously passed the Period Products Free Provision Bill of 2020, which mandated and provided funds to educational institutions and local authorities to provide free menstrual products to students and community members [8]. Both Scotland and the United States have set a precedent for the rest of the world by mandating free access to menstrual hygiene products through the allocation of government funds. 

Another pressing issue related to period poverty is that of taxation of menstrual products. In the United States, as of 20 September 2022, sales tax was still imposed on period products in 22 states, with Iowa and Virginia passing bills removing the tampon tax effective 1 January 2023 [10]. The Alliance for Period Supplies shared that, on average, menstruators require about 40 products per menstrual cycle [10]. Thus, the removal of the tampon tax will reduce the costs of menstrual products, increasing access to these essential hygiene products for those on a fixed income. On the federal level, while some progress has been made related to period poverty, barriers still exist [3,8]. For example, the Coronavirus Aid, Relief, and Economic Security Act expanded medical expense coverage to include period products [3]. However, the uninsured population was not eligible to use the benefit. Currently, federal governmental programs, like the Special Supplemental Nutrition Program for Women, Infants, and Children (WIC) and Supplement Nutrition Assistance Program (SNAP), do not allow funds to be used for the purchase of menstrual hygiene products in the United States [3]. Individuals participating in WIC and SNAP programs must undergo an application process and meet specific income and social guidelines [3]. By expanding the coverage of these programs to include the purchase of menstrual hygiene products, more individuals would have access to hygiene products, especially those with vulnerable financial and social circumstances. These programs are also federally monitored by the United States Department of Agriculture, which would universally expand access to menstrual products for WIC and SNAP participants in every state, with the rare exception of state waivers [8]. Globally, starting in 2004, countries such as Australia, Canada, Columbia, India, Kenya, Lebanon, Malaysia, Rwanda, and South Africa have removed any taxation related to period products [21]. 

Menstrual equity advocates have been working with legislators to require ingredient list labels on menstrual products and screenings for endometriosis, polycystic ovary syndrome, toxic shock syndrome, and other health conditions associated with menstrual product use [14]. Advocacy is also important for building capacity in the area of reproductive health education, which informs menstruators and the public about health risks and low-cost or no-cost grant-funded screenings for health conditions associated with menstrual hygiene products [14]. This type of proactive reform is also needed to increase accessibility to period products and empower safe menstrual hygiene management. 

Menstrual Management in a Global Context. Globally, in many provinces, territories, and countries, period poverty is still a major health issue that needs to be addressed. India is a global exemplar of period poverty. Media sources have shared that approximately 36 to 43 percent of the 355 million menstruating female population of India has not had access to menstrual products [22,23,24]. Thirty-six percent of females from over 35 cities in the country reported, in a survey, feeling uncomfortable purchasing them with other consumers around them [23]. While these statistics were alarming, even more concerning was the environment related to menses that women faced. The literature stressed that menstruating women were and had been historically considered impure and that discussion of a period was labeled a taboo topic by their society [25]. A 2020 report by the BBC noted that 71 percent of adolescent girls living in India were never educated on menstruation until it occurred personally to them [22]. Actively menstruating females have also reported being discriminated against by males who used shaming tactics, which led them to experience anxiety and fear, along with being barred from religious observance or social events during their menses [22]. 

One of the biggest barriers to period poverty in India is the affordability of sanitary products, especially for females living in rural areas [26]. Gupta shared that sanitary napkins are often seen as “faltu kharcha,” or an unnecessary expenditure, among people in these areas [24]. Instead, they continue to use rags and old pieces of clothing and rewash and hang these cloths to dry before use. There have been numerous attempts in India over the years to ramp up the ability to get products into the hands of those who need them and increase sanitary practices. These actions have occurred on many government levels, as well as by healthcare providers and non-governmental programs. Over a decade ago, a community development program, *Plan India*, spoke of the importance of hygienic practices and the use of products to reduce the risk of reproductive tract infections [2]. Since 2018, advances have been made in rural village areas related to production of biodegradable sanitary napkins, along with incense from flower waste [25,27,28]. Maharashtra, the second-most populous state in India, had this type of venture supported by the state government six years prior and led by local women self-help groups [27]. In addition to the products, education is provided related to maintaining health and hygiene during menstruation. This was deemed crucial to the project, as a prior health survey in a rural part of Maharashtra highlighted that menstruators between the ages of 11 and 19 years skipped academics for up to 60 days a year during their menstrual cycles [29]. Most recently, a site that manufactures biodegradable incense and sanitary napkins and is under the umbrella of the Pravara Institute of Medical Sciences of the University of Loni, Maharashtra, India, was visited twice by one of the authors [30]. The Institute supports *Pravara Jan Seva Mandal’s Women Empowerment Project*, which is led by women and addresses the menstrual hygiene needs of its village and neighboring rural area with the production and distribution of no- to low-cost products. While this initiative empowers the health of rural residents by increasing access to menstrual hygiene products, it simultaneously supports females in achieving financial independence through employment. This revolutionary initiative is capacity-building and sustainable within its current environment and provides a model for other communities to implement. It is recognized that there are many other organizations across the globe that are addressing period poverty. The non-governmental organization Women in International Security, founded in 1987, has recognized period poverty as a global crisis and recognized 10 groups or collections of constituents with the purpose of fighting for this cause. They include, but are not limited to, such organizations as th Alliance for Period Products, Freedom for Girls, Sustainable Health Enterprises, and The Pad Project [31].

## 3. Discussion

Menstrual and reproductive health issues are important for preserving the dignity and autonomy of women. Period poverty and its sequalae need to be addressed at all levels, and stakeholders should be encouraged to explore strategies that have appeared in the above literature review. These action plans and policies need to be adapted to be sustainable within specific rural or urban communities across the globe. At the individual level, menstruators need to feel empowered to speak up about the barriers they face related to menstrual health. Education programs need to start from a bottom-up approach led by recognized and respected laypersons, community health workers, and/or public health or healthcare professionals. Accurate information about menstrual hygiene practices needs to be conveyed to menstruators in low- and middle-income, hard-to-reach settings with vulnerable populations to dispel myths and misinformation, decrease cultural taboos, and improve personal reproductive system hygiene [5]. The male population living within these period poverty enclaves needs to be included in the educational process and made to recognize how their biases and negative beliefs about menstruation feed into gender inequality and discrimination. School systems can also take an active role related to period poverty by supporting individual and collective students of menstruating age by providing safe spaces for hygiene to occur, access to menstrual care products, and education related to the natural biological changes that occur in the female reproductive lifecycle, including menses, as called for in the literature [21,23]. Through education, negative social attitudes towards menstruation may lessen, and communities can catalyze social change that advocates for legislation addressing period poverty [5].

Governments need to take an active role in addressing period poverty. Legislation needs to be supported and enacted to create proactive change related to menstrual product affordability and availability. In the United States of America, advocates of women’s health need to lobby the United States Senate and House of Representatives to support the Menstrual Equity for All Act of 2023 to empower the health of the menstruating population [11,17]. Sustainable community resources, such as those used in the manufacture of biodegradable sanitary pads in India, can be shared and adopted by other countries with similar environmental and financial constraints. Whether appointed or elected, officials globally should be encouraged by their constituents to draft and support policy that allocates funding to educational institutions to supply students with free menstrual products in order to promote menstrual equity or period parity and allow menstruators to continue to attend school on a regular basis. Healthcare providers and public health workers, as stakeholders, can be called upon to work with respected key community members in all areas of the world to build capacity in the delivery of services related to menstrual health. Finally, research related to period poverty and its impact on global health efforts, particularly in vulnerable populations and financially challenged areas, needs to seek an increased stream of funding from government and private agencies devoted to women’s health. Dissemination of results must be widespread and shared with major organizations such as the World Health Organization and the UN so that programs and policies related to this global health issue can be developed and have far-reaching positive effects.

## 4. Conclusions

Period poverty is a gender and reproductive health issue that is influenced by various systemic and social factors and has real implications for the physical and psychosocial health of menstruators globally. Period poverty should be recognized as an urgent public health crisis. Everyone has a role in addressing period poverty by supporting officials that proactively advocate for legislation, policy, and programs that minimize barriers contributing to period poverty and promote menstrual equity. It is pivotal to recognize and address period poverty as a public health crisis to improve health outcomes on a global level, relative to SDGs three, four, five, six, and eight. These outcomes should include the right of the menstruating population to have their autonomy, dignity, health, and socio-emotional welfare respected and upheld.

## Figures and Tables

**Table 1 ijerph-20-07118-t001:** Summary of themes, key points, and relevance to public health.

Theme	Key Points	Relevance to Public Health
Overlooked Public Health Crisis	Period poverty continues to increase in prevalence and impact menstruators across the globe. Nearly half of the population is affected by menstruation. Yet, period poverty is not recognized or sufficiently discussed in clinical and legal spaces with the importance it deserves.	Period poverty is an immediate public health crisis that needs to be recognized and addressed by public health officials globally.
Repercussions of Period Poverty on Health, Wellness, and Education	Lack of access to menstrual hygiene products and washing facilities negatively impacts physiological, emotional, and social wellness. Menstruators encounter additional obstacles associated with poor menstruation management that prohibit attendance in academic and professional commitments, jeopardizing a menstruator’s ability to secure educational and financial independence.	Period poverty is a relevant and important health issue with real consequences on the social mobility of menstruators. Period poverty results in ineffective menstruation management methods that negatively impact physical and psycho-emotional health. Period poverty has direct consequences on sustainability and capacity building in the menstruating population presently and in the future. Initiatives that minimize poverty address the SDGs related to health and wellness, clean water and sanitation, education, and economic security.
Policy, Accessibility to Menstrual Hygiene Management, and Menstrual Equity Efforts	Globally, policy to increase access to menstrual hygiene products and washing facilities has been implemented by various nations. Federal and state mandates to supply menstruators with free menstrual hygiene products can alleviate the economic burden faced by menstruators. Taxation on menstrual products can impose additional financial barriers to maintaining proper menstrual hygiene. While access to menstrual management is a basic human right of menstruators, it is also a political issue that requires the use of policy to proactively address period poverty. Menstrual equity should be considered on political agendas.	Policy is a powerful tool in advancing menstrual equity agendas globally. Access to menstrual hygiene products needs to be considered on political agendas. Public health officials need to recognize the political implications of period poverty. Policy that increases access to menstrual hygiene management should be developed by legislators and supported by the public and elected officials.
Menstrual Management in a Global Context	Each culture and community have distinct beliefs regarding menstruation, with social and physical consequences that affect menstrual management. While some cultures may socially accept menstruation as an expected physiological process, others may view menstruation with a negative connotation and socially exclude menstruators, creating additional barriers to maintaining menstrual hygiene. *Pravara Jan Seva Mandal’s Women Empowerment Project* is an example of a sustainable community partner model in a rural area of India to provide access to menstrual hygiene products while empowering women to attain financial independence. There are various global organizations that address period poverty and its related initiatives to combat this issue. Four examples are provided in the manuscript, and others can be searched via social media.	Period poverty is a global issue that impacts every menstruator. Various cultural beliefs and stigmas regarding menstruation must be considered when designing, implementing, and evaluating initiatives addressing period poverty. The community partner model exemplified by Pravara Institute of Medical Science provides community stakeholders with a practical, sustainable model that increases access to menstrual hygiene products, empowers economic security amongst women, and promotes social mobility for women.

## Data Availability

Not applicable.

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
