# Peer review of "Period Poverty from a Public Health and Legislative Perspective"

_ijerph, 2023, doi:10.3390/ijerph20237118_

Round 1

Reviewer 1 Report

Comments and Suggestions for Authors

Period poverty and menstrual equity are gaining recognition as both a local and public health issue affecting women's health. This paper is of high significance and aligns well with the vision of the journal. However, certain issues should be addressed:

1. In the introduction, the authors referred to period poverty before defining the term. The last sentence of the in the first paragraph in the introduction should be included before referring to the term period poverty. 

2. In the Overlooked Public Health Crisis Section:

Please rephrase the sentence "The incarcerated menstruators are certain vulnerable population worth mention". 

3. Repercussions of period poverty: When you refer to the policies needed at the governmental level, it would be better to have a subsection on policies that is separate from the repercussions mentioned. 

4. Please restructure the first sentence in the menstrual equity efforts paragraph. It is worth "mentioning" rather than "mention"

5. Typo in Line 202 "Thirsty" rather than "Thirty"

6. It would be interesting to also talk about refugees and immigrants struggling with this issue. There are some papers on menstrual health in refugee camps as they are severely disproportionately impacted.

7. Another interesting area to focus on is the work of international organizations on addressing this issue in vulnerable areas around the world in comparison to what is being done in the U.S. by local NGOs.

8. For the methods, can you elaborate more on why you only selected Google scholar? Is this a narrative review?

Comments on the Quality of English Language

Appropriate, but can be improved.

Author Response

Reviewer 1:

  1. In the introduction, the authors referred to period poverty before defining the term. The last sentence of the in the first paragraph in the introduction should be included before referring to the term period poverty.

Recommendation accepted. Period poverty definition provided before using the term.

  1. In the Overlooked Public Health Crisis Section:
    Please rephrase the sentence "The incarcerated menstruators are a certain vulnerable population worth mentioning".

Recommendation accepted. Sentence removed and rephrased.

  1. Repercussions of period poverty:
    When you refer to the policies needed at the governmental level, it would be better to have a subsection on policies that is separate from the repercussions mentioned.

Recommendation accepted. Moved text to policy theme section.

  1. Please restructure the first sentence in the menstrual equity efforts paragraph. It is worth "mentioning" rather than "mention"

Recommendation accepted.

  1. Typo in Line 202 "Thirsty" rather than "Thirty"

Recommendation accepted.

  1. It would be interesting to also talk about refugees and immigrants struggling with this issue. There are some papers on menstrual health in refugee camps as they are severely disproportionately impacted.

Recommendation accepted. Clarified content addressing impact of period poverty on migrant/displaced populations briefly. This review paper attempts to highlight the significance of period poverty and provide a brief overview of the issue, hence specific issues within this issue area are addressed briefly.

  1. Another interesting area to focus on is the work of international organizations on addressing this issue in vulnerable areas around the world in comparison to what is being done in the U.S. by local NGOs.

Recommendation accepted. Added sample of global NGOs to manuscript (lines 258-263).

  1. For the methods, can you elaborate more on why you only selected Google scholar? Is this a narrative review?

Recommendation accepted. Answered in the section of manuscript related to search engines/search criteria (lines 62-68)

Reviewer 2 Report

Comments and Suggestions for Authors

Thank you for providing an opportunity to review this manuscript: Period Poverty from a Public Health and Legislative Perspective. In fact, the manuscript stands between a commentary and a review on period poverty and its relevance with sustainable goals and also public health. Overall I have found the paper interesting. However, I think the manuscript could be improved in the following areas:

1. Please add a definition for Period poverty either before the Literature review or right at the beginning of the literature review.

2. Please provide a table to summarize the literature based on your interest. 

I congratulate the authors for writing this work.

Author Response

Reviewer 2:

  1. Please add a definition for Period poverty either before the Literature review or right at the beginning of the literature review.

Recommendation accepted. Definition of period poverty provided very clearly in the introduction.

  1. Please provide a table to summarize the literature based on your interest.

Recommendation accepted. Please review the table provided highlighting the themes/main ideas from review, key points, and recommendations for public health for each theme. This is placed after the References for editors to insert where they see fit.

Reviewer 3 Report

Comments and Suggestions for Authors

This paper addresses period poverty, which is an important and timely public health issue.

Is this paper supposed to be a global examination of period poverty, or just focusing on the US?

Introduction – Provide a brief explanation about the Sustainable Development Goals (what they are, why they are important).

Check for appropriate punctuation use (e.g. use of semi-colon when a colon should be used.

Literature review:

Were the authors meaning this to be a meta-analysis?

Why only search for literature from 2018-2023?

Why not include additional terminology such as menstruation hygiene, menstruation management, etc?  The use of only “period poverty” as a search term is very limiting.

Authors limited their search to female gender. More research is coming out related to people who menstruate, not limited by gender. Authors should consider this as an issue moving forward.

What specific age-related qualifiers were used?  And why limit by age, as individuals are menstruating earlier and not enter menopause until later?

Did you search just for “period poverty” when you were seeking out organizations and programs?  Did you include ALL articles and website from your review, or what were your inclusion criteria?  What type of analysis was conducted to develop the four themes?  Authors should provide additional information about their methodology.

Line 60 – Should be “glean” not “gleam.”

Line 73-74 – I am confused why authors are focusing on New Jersey.

On line 67, authors mention that 2 in 5 women struggle to purchase menstrual hygiene products, but then in line 71, authors mention that 1 in 6 women between 12 and 44 below the poverty line struggles with period poverty.  Maybe differentiate more the differences in the statistics provided here.

Line 78-79 – not sure this information is needed as information related to income and period poverty has already been provided. The information provided here is just conjecture.

The paragraph on incarcerated menstruators comes out of nowhere – while an important issue, I am not sure why this is included, or included to the extent that it is?  A focus on adolescent period poverty would be more beneficial, or period poverty across the lifespan, menstrual cycle.

As there are separate sections for the other three themes, there should also be a separate section for the theme entitled “Addressing Period Poverty at the Individual, Governmental, Global or Societal Level”.

I am not sure how these two themes are different: ‘Overlooked PH Crisis’ and ‘Repercussions of Period Poverty’, as authors discuss the epidemiology of period poverty in both sections. The information related to adolescent period poverty should be moved to ‘Overlooked PH Crisis.’

Comments on the Quality of English Language

Minor English language and punctuation edits should be made.

Author Response

Reviewer 3:

  1. Is this paper supposed to be a global examination of period poverty, or just focusing on the US?

This paper includes both components related to period poverty. As the authors are from the U.S. and the second author practices in women's health in the U.S. the topic has become a major interest. This interest was sparked by her academic consultation and primary care work with the Pravara Institute of Medical Sciences in India over the past 4 years. Examples are shared in both of these countries as well as examples of other countries either related to the issue or policies addressing it.

  1. Introduction – Provide a brief explanation about the Sustainable Development Goals (what they are, why they are important).

Recommendation accepted. Importance of the Sustainable Development Goals described and each goal is briefly explained. The goal of this manuscript is to highlight the significance of period poverty and other dimensions of this social-health issue. Although the Sustainable Development Goals are relevant in this context, they are not the main topic of the paper, and hence, the Sustainable Development Goals are not explained extensively. 

  1. Check for appropriate punctuation use (e.g. use of semi-colon when a colon should be used.

Recommendation accepted.

4. Literature review:
a. Were the authors meaning this to be a meta-analysis?

It is not a meta-analysis.  The term literature review was used but its intent is to provide background information and highlight current information on the topic of period poverty.  This is now explained in the manuscript.

b. Why only search for literature from 2018-2023?

The authors wanted the literature to be current and thus chose a five-year time frame for the search vs. extended back in time.

c. Why not include additional terminology such as menstruation hygiene, menstruation management, etc.? The use of only “period poverty” as a search term is very limiting.

The suggested search terms were added as per recommendation. 

d. Authors limited their search to female gender. More research is coming out related to people who menstruate, not limited by gender. Authors should consider this as an issue moving forward.

This recommendation is taken under consideration for future research and review of the topic.

e. What specific age-related qualifiers were used?

Adolescence to adult menstruators (see lines 65-68). 

f. And why limit by age, as individuals are menstruating earlier and not enter menopause until later?

Resolved.

g. Did you search just for “period poverty” when you were seeking out organizations and programs?

See updated search terms in section of keywords and in collusion criteria (lines 23-24 & 64-73)  

h. Did you include ALL articles and website from your review, or what were your inclusion criteria? What type of analysis was conducted to develop the four themes?  Authors should provide additional information about their methodology.

See above comment.

  1. Line 60 – Should be “glean” not “gleam.”

Recommendation accepted.

  1. Line 73-74 – I am confused why authors are focusing on New Jersey.

Recommendation accepted. This information providing statistical data on the prevalence of period poverty in New Jersey was removed.

  1. On line 67, authors mention that 2 in 5 women struggle to purchase menstrual hygiene products, but then in line 71, authors mention that 1 in 6 women between 12 and 44 below the poverty line struggles with period poverty. Maybe differentiate more the differences in the statistics provided here.

Recommendation accepted. Highlighted how 2 in 5 women struggle to purchase menstrual hygiene products at the national level compared to those women below the poverty line. Clarified the significance of the federal poverty line in relation to women experiencing period poverty.

  1. Line 78-79 – not sure this information is needed as information related to income and period poverty has already been provided. The information provided here is just conjecture.

Recommendation accepted. Rephrased and revised section regarding period poverty in the context of socio-economics and the importance of considering economics in discussion of period poverty.

  1. The paragraph on incarcerated menstruators comes out of nowhere – while an important issue, I am not sure why this is included, or included to the extent that it is? A focus on adolescent period poverty would be more beneficial, or period poverty across the lifespan, menstrual cycle.

Recommendation considered. Thank you for your comment. This specific section focuses on briefly highlighting specific populations that are often excluded from conversations regarding period poverty and are more vulnerable to experiencing period poverty. This section has been edited to include migrant/displaced populations and include context on the relevance of this information.

  1. As there are separate sections for the other three themes, there should also be a separate section for the theme entitled “Addressing Period Poverty at the Individual, Governmental, Global or Societal Level”.

Recommendation accepted. Separate section created for the fourth theme which is changed to “Policy, Accessibility to Menstrual Hygiene Management, and Menstrual Equity Efforts.”

  1. I am not sure how these two themes are different: ‘Overlooked PH Crisis’ and ‘Repercussions of Period Poverty’, as authors discuss the epidemiology of period poverty in both sections. The information related to adolescent period poverty should be moved to ‘Overlooked PH Crisis.’

Recommendation accepted. These two themes are separate. While the “Overlooked PH crisis” theme focuses on the prevalence and growing concern of period poverty from an epidemiologic perspective, the “Repercussions of Period Poverty” theme refers to the negative effects of poverty on health, wellness, education, and social mobility. The “Repercussions of Period Poverty” theme is reworded to “Repercussions of Period Poverty on Health, Wellness, and Education” theme to further specify the content.

Round 2

Reviewer 1 Report

Comments and Suggestions for Authors

We thank the authors for making the recommended changes. This topic is of high importance, and more emphasis should be provided in the literature regarding the impact of this topic on women's health.